# Understanding Catastrophic Overfitting in Fast Adversarial Training From a Non-robust Feature Perspective

## Abstract

To make adversarial training (AT) computationally efficient, FGSM AT has attracted significant attention. The fast speed, however, is achieved at the cost of catastrophic overfitting (CO), whose reason remains unclear. Prior works mainly study the phenomenon of a significant PGD accuracy (Acc) drop to understand CO while paying less attention to its FGSM Acc. We highlight an intriguing CO phenomenon that FGSM Acc is higher than accuracy on clean samples and attempt to apply non-robust feature (NRF) to understand it. Our investigation of CO by extending the existing NRF into fine-grained categorization suggests: there exists a certain type of NRF whose usefulness is increased after FGSM attack, and CO in FGSM AT can be seen as a dynamic process of learning such NRF. Therefore, the key to preventing CO lies in reducing its usefulness under FGSM AT, which sheds new light on understanding the success of a SOTA technique for mitigating CO.

## 1 Introduction

Despite impressive performance, deep neural networks (DNNs) (LeCun et al., 2015; He et al., 2016; Huang et al., 2017; Zhang et al., 2019a; 2021) are widely recognized to be vulnerable to adversarial examples (Szegedy et al., 2013; Biggio et al., 2013; Akhtar & Mian, 2018). Without giving a false sense of robustness against adversarial attacks (Carlini & Wagner, 2017; Athalye et al., 2018; Croce & Hein, 2020), adversarial training (AT) (Madry et al., 2018; Zhang et al., 2019c) has become the de facto standard approach for obtaining an adversarially robust model via solving a min-max problem in two-step manner. Specifically, it first generates adversarial examples by maximizing the loss, then trains the model on the generated adversarial examples by minimizing the loss. PGD-$N$ AT (Madry et al., 2018; Zhang et al., 2019c) is a classical AT method, where $N$ is the iteration steps when generating the adversarial samples in inner maximization. Notably, PGD-$N$ AT is $N$ times slower than its counterpart standard training with clean samples. A straightforward approach to make AT faster is to set $N$ to 1, *i.e* reducing the attack in the inner maximization from multi-step PGD to single-step FGSM (Goodfellow et al., 2015). For simplicity, PGD-based AT and FGSM-based fast AT are termed PGD AT and FGSM AT, respectively.

FGSM AT often fails with a sudden robustness drop against PGD attack while maintaining its robustness against FGSM attack, which is called catastrophic overfitting (CO) (Wong et al., 2020). With Standard Acc denoting the accuracy on clean samples while FGSM Acc and PGD Acc indicating the accuracy under FGSM and PGD attack, we emphasize that a CO model is characterized by two main phenomena as follows.

- Phenomenon 1: The PGD Acc drops to a value close to zero when CO happens (Wong et al., 2020; Andriushchenko & Flammarion, 2020).
- Phenomenon 2: FGSM Acc is higher than Standard Acc for a CO model (Kim et al., 2020; Andriushchenko & Flammarion, 2020).

Multiple works (Wong et al., 2020; Kim et al., 2020; Andriushchenko & Flammarion, 2020) have focused on understanding CO by explaining the drop of PGD Acc in Phenomenon 1; however, they pay less attention to Phenomenon 2 regarding FGSM Acc. Specifically for Phenomenon 1, FGSM-RS (Wong et al., 2020) attributes it to the lack of perturbation diversity in FGSM AT, which

is refuted by a follow-up GradAlign (Andriushchenko & Flammarion, 2020) by demonstrating a co-occurrence of local non-linearity and the PGD Acc drop. However, these understandings cannot explain why FGSM Acc is higher than Standard Acc for a CO model in Phenomenon 2.

In the context of adversarial learning, numerous works (Goodfellow et al., 2015; Tabacof & Valle, 2016; Tanay & Griffin, 2016; Koh & Liang, 2017; Nakkiran, 2019; Athalye et al., 2018; Zhang et al., 2020) have attempted to explain why adversarial examples exist from different angles, among which non-robust feature (NRF) (Ilyas et al., 2019) is a popular one which also aligns well with all other explanations (Goodfellow et al., 2015; Tabacof & Valle, 2016; Tanay & Griffin, 2016; Koh & Liang, 2017; Nakkiran, 2019; Athalye et al., 2018). Such compatibility suggests that the NRF perspective constitutes an essential tool for understanding adversarial vulnerability, to which CO is also directly related. Specifically, the authors of (Ilyas et al., 2019) define the positive-correlation between features and true labels as *feature usefulness* (see Section 3.1 for more detailed definitions). Therefore, the adversarial vulnerability of DNNs is attributed to the existence of non-robust features (NRFs), which can be made anti-correlated with the true label under adversary. This understanding of NRFs in (Ilyas et al., 2019) well aligns with the fact that a CO model achieves close to zero robustness against PGD attack, and thus motivates us to believe that the NRF perspective might be an auspicious direction for understanding CO in FGSM AT.

The NRF in (Ilyas et al., 2019) is defined with PGD attack, which is followed in this work; however, we extend their NRF framework by additionally considering FGSM attack for fine-grained categorization. Considering the difference of adversarial attack strength between FGSM and PGD attack, GradAlign (Andriushchenko & Flammarion, 2020) explains Phenomenon 1 by demonstrating how well the attack variant (FGSM or PGD attack) can solve the inner maximization problem in AT. We start our investigation by providing an alternative interpretation of this adversarial strength difference between the two attack variants within the NRF framework (Ilyas et al., 2019), named **strength-based NRF categorization**. Despite aligning well with Phenomenon 1, We find that this strength-based categorization cannot explain Phenomenon 2 since the usefulness of these NRFs is *decreased* under FGSM attack and leads to an *decrease* (instead of *increase* in Phenomenon 2) of classification accuracy on FGSM adversarial examples than clean samples.

To understand Phenomenon 2 in CO from the NRF perspective, we conjecture that there exists a type of NRF whose usefulness is *increased* under FGSM attack, thus can lead to a higher FGSM Acc than Standard Acc (Phenomenon 2). In other words, if such type of NRFs (NRF2 in the following categorization) exists, Phenomenon 2 can be justified. Considering whether the usefulness is *decreased* or *increased* under FGSM attack, we propose a **direction-based NRF categorization** where NRF2 (NRF1) leads to the *increase* (*decrease*) of classification accuracy under FGSM attack. To prove the existence of NRF2, we follow the procedure of verifying the existence of NRF in (Ilyas et al., 2019). Moreover, we show that NRF2 can cause a significant PGD Acc drop , which also helps justify Phenomenon 1 in CO.

Overall, towards understanding CO in FGSM AT, our contributions are summarized as follows:

- Our work shifts the previous focus on PGD Acc in Phenomenon 1 to FGSM Acc in Phenomenon 2 for understanding CO. Given NRF as a popular perspective on adversarial vulnerability, we are the first to attempt at applying it to explain Phenomenon 2.

- We extend the existing NRF framework under PGD attack (Ilyas et al., 2019) to more fine-grained NRF categorization by FGSM attack. We verify the existence of NRF2 and show that its existence well justifies Phenomenon 2 (as well as Phenomenon 1).

- Very recent works show that adding noise on the image input achieves SOTA performance for FGSM AT. However, their mechanism of such a simple technique preventing CO remains not fully clear, for which our NRF2 perspective shed new light on its success.

## 2 PROBLEM OVERVIEW AND RELATED WORK

### 2.1 FGSM AT AND EXPERIMENTAL SETUPS

Let $\mathcal{D}$ denote a data distribution with $(x, y)$ pairs and $f(\cdot, \theta)$ parameterized by $\theta$ denote a deep model. For standard training, the model $f(\cdot, \theta)$ is trained on $\mathcal{D}$ by minimizing $\mathbb{E}_{(x,y)\sim\mathcal{D}}[l(f(x, \theta), y)]$, where $l$ indicates a cross-entropy loss for a typical multi-class classification task. Adversarial training

(AT) (Madry et al., 2018) for obtaining a robust model is formalized as a min-max optimization problem:

$$\arg\min_{\theta} \mathbb{E}_{(x,y)\sim\mathcal{D}} \left[ \max_{\delta\in\mathbb{S}} l(f(x+\delta;\theta), y) \right], \tag{1}$$

where $\mathbb{S}$ is a perturbation limitation ($\epsilon$ with the $l_\infty$ constraint in this work). The outer minimization problem in AT is often the same as standard training; however, AT has an unique inner maximization problem that seeks a perturbation inside the $\mathbb{S}$ for maximizing the optimization loss. PGD AT and FGSM AT are two typical adversarial training methods with PGD attack and FGSM attack solving the inner maximization problem, respectively.

**Experimental setups.** Unless specified, we follow the settings in GradAlign (Andriushchenko & Flammarion, 2020) during training and evaluation. The experiments are conducted on CIFAR10 with PreAct ResNet-18, trained for 30 epochs with cyclic learning rates and half-precision training. We adopt SGD optimizer with weight decay $5 \times 10^{-4}$, and the maximum learning rate is set to 0.2. $\ell_\infty$ attack with perturbation constraint $\epsilon$=8/255 is applied in both training and evaluation. Following (Wong et al., 2020; Andriushchenko & Flammarion, 2020), we calculate the Standard accuracy (Standard Acc) on clean samples, FGSM accuracy (FGSM Acc), and PGD accuracy (PGD Acc) under PGD-50-10 attack (performing PGD-50 attack with ten restarts and step size $\alpha = \epsilon/4$) for evaluation.

## 2.2 CATASTROPHIC OVERFITTING IN FGSM AT

**What are the CO Phenomena?** Notably, a model trained only on adversarial examples generated by FGSM attack in FGSM AT still has robustness against PGD attack. In practice, this robustness is only slightly lower than that of much more computationally expensive PGD AT. However, this robustness level can often not be maintained till the end of training as a classical PGD AT. Specifically, as the FGSM AT evolves, the model robustness against PGD attack first increases but then enters a phase where the robustness quickly drops to and stays at zero. Following (Wong et al., 2020), this phase is termed catastrophic overfitting (CO). Another intriguing phenomenon related to CO is that for a model at the phase of CO, it achieves a higher FGSM Acc than Standard Acc (Kim et al., 2020; Andriushchenko & Flammarion, 2020). We term these two phenomena regarding CO as Phenomenon 1 and Phenomenon 2 respectively, as in Section 1.

**How to explain the CO phenomena?** With the finding that random initialization of perturbation helps alleviate CO (Wong et al., 2020), a tempting explanation suggests that the CO in FGSM AT lies in the lack of perturbation diversity, which has been refuted by (Andriushchenko & Flammarion, 2020). Instead, it attributes the reason for the PGD Acc drop to local non-linearity, which is quantified by the *gradient alignment*: $\cos(\nabla_x\ell(x, y; \theta), \nabla_x\ell(x+\eta, y; \theta))$. The local non-linearity (low gradient alignment) indicates a low linear approximation quality of FGSM perturbations to PGD perturbations. In other words, local non-linearity means that the inner maximization problem in Eq 1 cannot be solved accurately by FGSM. It is demonstrated in (Andriushchenko & Flammarion, 2020) that local linearity decreases significantly when CO happens in FGSM AT. Their perspective is mainly dependent on the co-occurrence between non-linearity and the drop of PGD Acc. In other words, the non-linearity perspective exclusively focuses on explaining Phenomenon 1, for which this work provides an alternative NRF explanation (see Section 3). More importantly, our work fills the gap to explain Phenomenon 2 from a NRF perspective (see Section 4).

**How to prevent CO?** With the focus on Phenomenon 1, numerous works have attempted to prevent CO. Fast AT (Wong et al., 2020) is the first to show FGSM AT can achieve comparable robustness as PGD AT of "free" variants (Shafahi et al., 2019; Zhang et al., 2019b). A follow-up work (Andriushchenko & Flammarion, 2020) shows that CO still occurs in (Wong et al., 2020) when the step size increases and introduces a regularization loss (GradAlign) for maximizing local linearity to avoid CO. Other successful attempts for avoiding CO include adaptive perturbation size (Kim et al., 2020), dynamic dropout scheduling (Vivek & Babu, 2020) and detection-based alternating strategy (Li et al., 2020). Intriguingly, very recent works (Zhang et al., 2022; de Jorge et al., 2022) have shown that adding noise on the image input is sufficient for preventing collapse and achieves SOTA performance. However, the reason for its success remains not fully clear, for which our NRF perspective with direction-based categorization provides an explanation (see Section 5).

## 3 NON-ROBUST FEATURE PERSPECTIVE ON ADVERSARIAL TRAINING

Before investigating CO from the NRF perspective, we first revisit the definition and methodology of robust and non-robust features defined in (Ilyas et al., 2019) (Fig. 1(a)). Considering the difference of attack strength between FGSM attack and PGD attack, we extend the non-robust features defined in (Ilyas et al., 2019) to a fine-grained categorization under FGSM attack (strength-based categorization in Fig. 1(b)) and discuss its relationship with CO phenomena.

### 3.1 BACKGROUND ON FEATURE USEFULNESS AND ROBUSTNESS

Here, we revisit the definitions and methodology of DNN features introduced in (Ilyas et al., 2019). According to (Ilyas et al., 2019), a *feature* is defined as a function mapping from the input space $\mathcal{X}$ to real numbers, *i.e* $f : \mathcal{X} \to \mathbb{R}$, where $\mathbb{R}$ can be the label space in classification task. Therefore, a DNN classifier can be perceived as a function utilizing a set of *useful features* for label prediction (Ilyas et al., 2019), where *useful features* in (Ilyas et al., 2019) are characterized by their positive correlation with true label, defined as:

- **$\rho$-useful features:** A feature $f$ is $\rho$-useful ($\rho > 0$) if it is correlated with the true label in expectation, shown as follows:

$$\mathbb{E}_{(x,y)\sim\mathcal{D}}[y \cdot f(x)] \geq \rho. \tag{2}$$

To understand adversarial vulnerability, (Ilyas et al., 2019) further proposes to dichotomize the above useful features into robust features (RFs) and non-robust features (NRFs), defined as follows:

- **Robust feature (RFs):** a useful feature $f$ is robust if there exists a $\gamma > 0$ for it to be $\gamma$-robustly useful under some specified set of valid perturbations $\Delta$, shown as follows:

$$\mathbb{E}_{(x,y)\sim\mathcal{D}}[\inf_{\delta\in\Delta(x)} y \cdot f(x + \delta)] \geq \gamma. \tag{3}$$

- **Non-robust feature (NRFs):** a useful feature $f$ is non-robust if $\gamma > 0$ does not exist.

**Adversarial vulnerability can be attributed to the existence of NRFs (Ilyas et al., 2019).** As discussed in (Ilyas et al., 2019), adversarial vulnerability is caused by the presence of NRFs which are useful and predictive. According to (Ilyas et al., 2019), *"in the presence of an adversary, any useful but non-robust features can be made **anti-correlated** with the true label, leading to adversarial vulnerability"* (Ilyas et al., 2019). Therefore, adversarial training obtains a robust model by discouraging from learning NRFs. In practice, finding a worst-case perturbation under a certain budget for Eq. 3 is not feasible since it is often an NP-hard problem (Katz et al., 2017; Weng et al., 2018), and thus (Ilyas et al., 2019) uses multi-step PGD attack to approximate such a worst-case solution when investigating NRFs.

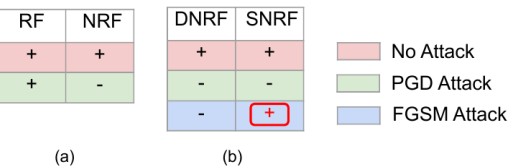

Figure 1: Strength-based NRFs categorization. (a) Definitions of RFs and NRFs in (Ilyas et al., 2019), where the plus($+$)/minus($-$) sign indicate that the features are *positive-correlated* or *anti-correlated* with true label, respectively. (b) Definitions of DNRF. Considering the attack strength, DNRF are made *anti-correlated* by both FGSM and PGD attack, while SNRF is made *anti-correlated* by PGD attack but still *positive-correlated* with true labels under FGSM attack.

Fig. 1 (a) summarizes the feature definition in (Ilyas et al., 2019). Specifically, the plus sign ($+$) indicates the useful features which has *positive* correlation with true labels, while the minus sign ($-$) indicates *anti-correlated* features under PGD attack.

**Verifying the existence of NRFs (Ilyas et al., 2019).** The procedure verifying the existence of NRFs in (Ilyas et al., 2019) is summarized in Fig. 2(a) by three steps. At Step 1, it trains a model $\mathcal{M}_1$ with standard training on the original training set $(X_{train}, y)$, where $X_{train}$ and $y$ indicate the training sample and its corresponding true label, respectively. At Step 2, it first randomly picks a random label $y_{rand}$ for each training sample to ensure that the training set $X_{train}$ has no features with a positive correlation with the random label $y_{rand}$. After that, perturbation $\delta$ is generated by PGD attack on $\mathcal{M}_1$ by making sample prediction $f(x + \delta)$ *close to* $y_{rand}$. This step aims to generate a perturbation $\delta$

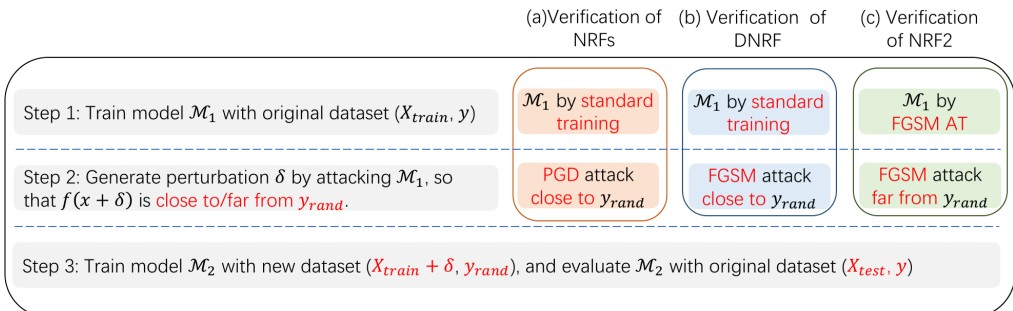

Figure 2: Experiment procedures for verifying the existence of non-robust features with three basic steps. (a) verifies the existence of NRF (mainly NRF1) in (Ilyas et al., 2019); (b) verifies the existence of DNRF; and (c) verifies the existence of NRF2.

which includes NRFs related to $y_{rand}$. At Step 3, model $\mathcal{M}_2$ is trained on the new dataset ($X_{train}+\delta$, $y_{rand}$) generated at Step 2, and then evaluated on the original test dataset with true labels ($X_{test}$, $y$). According to (Ilyas et al., 2019), the perturbation $\delta$ is the only connection between $X_{train}+\delta$ and $y_{rand}$ since there is no positive correlation between $X_{train}$ and $y_{rand}$. Therefore, if model $\mathcal{M}_2$ achieves higher accuracy than random prediction (*e.g* 10% for CIFAR10) on the original test dataset ($X_{test}$, $y$) with true labels, the existence of NRFs in $\delta$ is verified. We re-implement this experiment in (Ilyas et al., 2019), and $\mathcal{M}_2$ achieves a accuracy of 48.16% (with five independent runs), as shown in Table 1, which verifies the existence of NRFs as in (Ilyas et al., 2019).

## 3.2 STRENGTH-BASED NRF CATEGORIZATION

It is widely known that PGD attack is stronger than FGSM attack, which is supported by the finding that FGSM Acc is higher than PGD Acc under the same $l_\infty$ perturbation budget (Madry et al., 2018). Thus, PGD Acc is often adopted as a common metric to evaluate the model robustness. FGSM AT is faster than PGD AT but at the cost of a mildly lower PGD Acc (than PGD AT) even when CO does not happen in FGSM AT. When CO occurs, the PGD Acc drops to a value close to zero (Phenomenon 1). Since the difference between PGD AT and FGSM AT lies in the attack variant, GradAlign (Andriushchenko & Flammarion, 2020) explains their difference based on how well the adopted attack can solve the inner maximization problem. Specifically, FGSM AT yields lower robustness because FGSM attack cannot solve the problem as accurately as PGD attack because PGD attack is stronger than its FGSM counterpart. The following discussion provides an alternative interpretation of the attack strength-based explanation in (Andriushchenko & Flammarion, 2020) from the NRF perspective.

**Intuitive categorization.** Considering the attack strength difference, the NRFs can be divided into two types, as shown in Figure 1(b). The first type of NRFs is named as *double* non-robust feature (DNRF) since it can be made anti-correlated with the true labels by both FGSM and PGD attack. The existence of DNRF explains why FGSM AT yields a more robust model than standard training against PGD attack during evaluation. By contrast, the other type of NRFs is called *single* non-robust feature (SNRF) since it is made anti-correlated with true labels by PGD attack but is still positive-correlated with true labels under FGSM attack.

**Experimental verification of DNRF.** This setup follows the procedure in (Ilyas et al., 2019) (Fig. 2(a)) with a small modification. With its definition, DNRF has the property of being made anti-correlated with true labels under both PGD attack and FGSM attack. Therefore, the existence of DNRF ensures that the test acc will

Table 1: Verifying the existence of DNRF. The experimental procedure follows Fig. 2(a) and (b), with PGD and FGSM attack at Step 2 respectively.

| Features | Attack at Step 2 | Test Acc of $\mathcal{M}_2$ |
|---|---|---|
| NRFs (Ilyas et al., 2019) | PGD (Fig. 2(a)) | 48.16±5.12 |
| DNRF | FGSM (Fig. 2(b)) | 20.01±1.16 |

also be higher than random guess (10% for CIFAR10) if we replace the PGD attack at Step 2 with FGSM attack, as shown in Fig. 2(b). This is confirmed by an accuracy of 20.01% on the original test set, see Table 1.

**On SNRF and its relationship with CO phenomena.** It is challenging to directly verify the existence of SNRF. The phenomenon that 20.01% (FGSM attack) is lower than 48.61% (PGD attack) in Table 1 can be seen as an indirect evidence for the existence of SNRFs which can be extracted by PGD attack but not FGSM attack. Even though direct empirical verification of SNRF is challenging, its theoretical existence is straightforward as long as FGSM attack is weaker than PGD attack. Moreover, the weaker the FGSM attack (compared with PGD attack), the more SNRF. FGSM AT cannot effectively discourage the model from learning SNRF as PGD AT, and thus we can attribute the lower PGD Acc of FGSM AT than PGD AT to the existence of SNRF. However, SNRF under strength-based NRF categorization might (at most) partly explain CO Phenomenon 1 but cannot justify CO Phenomenon 2. The reason is that the model might have a very low PGD Acc, but FGSM Acc cannot be higher than Standard Acc even in an extreme case when all the NRFs become SNRF due to a very weak FGSM attack. The following section introduces a new NRF categorization to better explain CO phenomena, especially Phenomenon 2.

# 4 DIRECTION-BASED NRF CATEGORIZATION FOR UNDERSTANDING CO PHENOMENA

**Direction-based NRF categorization.** Similar to the above strength-based NRF categorization, the categorization here considers FGSM attack but differs by a key assumption: whether the usefulness of certain NRFs is *decreased* or *increased* under FGSM attack. We call this NRF categorization as direction-based, which is defined as follows:

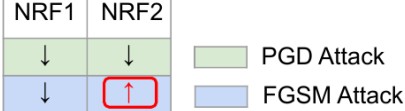

Figure 3: Change of usefulness under attack for NRF1 and NRF2. The $\uparrow$/$\downarrow$ indicate the increase/decrease of feature usefulness under attack, respectively.

- **NRF1:** NRF1 is a type of NRF whose usefulness is *decreased* after FGSM attack, thus can be exploited by FGSM attack to *decrease* the classification accuracy after FGSM attack.

- **NRF2:** NRF2 is a type of NRF whose usefulness is *increased* after FGSM attack, thus can be exploited by FGSM attack to *increase* the classification accuracy after FGSM attack.

NRF1 and NRF2 still follow the definition of NRF regarding PGD attack. In other words, the usefulness of both NRF1 and NRF2 is decreased after PGD attack. The change of their usefulness after attacks is summarized in Fig. 3, where the increase and decrease of feature usefulness are denoted by the $\uparrow$ and $\downarrow$, respectively.

## 4.1 ON NRF2 EXISTENCE AND ITS EXPLANATION FOR PHENOMENON 2

When we discuss DNRF and SNRF in Section 3.2, by default, we assume that their usefulness is decreased after FGSM attack, and thus they can be seen as NRF1. In other words, the existence of NRF1 is straightforward; however, it is unclear whether NRF2 actually exists.

*Conjecture 1*: We conjecture that there exists NRF2, and the FGSM attack in AT encourages the model to learn NRF2.

**Differences between verifying NRF1 and NRF2.** The experimental procedure of verifying NRF2 is shown in Fig. 2(c). The key reason why procedures in Fig. 2 can verify the existence of certain NRFs is that the generated perturbation $\delta$ is the only connection between $X_{train} + \delta$ and $y_{rand}$, and it should include certain NRFs related to $y_{rand}$. In other words, the usefulness of certain NRFs should be *increased* after attack at Step 2 of Fig. 2. For FGSM attack, $f(x + \delta)$ is optimized to be *far from* the true label $y$ by maximizing the loss $l(f(x + \delta), y)$, and the usefulness of NRF1 and NRF2 are *decreased* and *increased* by definition, respectively(see Fig. 3). Therefore, to *increase* the usefulness of NRF1 at Step 2, the optimization goal should be *close to* $y_{rand}$, as shown in Fig. 2(b). By contrast, to verify the existence of NRF2, the optimization goal at Step 2 should follow that of FGSM attack, *i.e far from* $y_{rand}$, as shown in Fig. 2(c), which *increases* the usefulness of NRF2.

**Verification of Conjecture 1**. As discussed above, verifying the existence of NRF2 requires an opposite optimization goal with that of NRF1 at Step 2 (see Fig. 2(c)). For the model $\mathcal{M}_1$ at Step 1, we adopt FGSM AT with the results reported in Table 2. When $\mathcal{M}_1$ at Step 1 is set to a CO model with FGSM AT, our model $\mathcal{M}_2$ evaluated on the original test set achieves an accuracy of around

$17.52\% \pm 1.59\%$ (with five independent runs), which verifies the existence of NRF2 since it is higher than 10% (random prediction for CIFAR10). Given that the usefulness of NRF2 is increased after FGSM attack, the existence of NRF2 in a model after CO justifies why FGSM Acc can be higher than Standard Acc (CO Phenomenon 2). Moreover, we experiment with setting the $\mathcal{M}_1$ at Step 1 to one with standard training, the Test Acc of $\mathcal{M}_2$ is close to random prediction, suggesting that it is the FGSM attack in AT that encourages the model to learn NRF2.

**Does NRF2 exist in a model before CO in FGSM AT?** It is interesting to investigate whether NRF2 exists for the FGSM AT model before CO. To this end, we set $\mathcal{M}_1$ at Step 1 of Fig. 2(c) to a FGSM AT model saved before CO, which yields an $\mathcal{M}_2$ with an accuracy close to random prediction(see FGSM AT(Before CO) in Table 2). This indicates that NRF2 mainly exists in the model after CO in FGSM AT, which further confirms the relationship between CO and NRF2.

Table 2: Verifying the existence of NRF2 as shown in Figure 2(c). Ablation studies of different $\mathcal{M}_1$ at Step 1 are also implemented.

| Model $\mathcal{M}_1$ | Test Acc of $\mathcal{M}_2$ |
|---|---|
| Standard Training | 10.42±0.74 |
| FGSM AT (After CO) | **17.52±1.59** |
| FGSM AT (Before CO) | 9.48±2.01 |

**Can NRF2 be exploited by FGSM attack to decrease FGSM accuracy?** (Kim et al., 2020) reports that FGSM Acc is higher than Standard Acc when CO happens in FGSM AT. Here, we show that this is not always the case if we evaluate FGSM Acc of the CO model with different step sizes, as shown in Table 3. Note that the result with step size of zero indicates the Standard Acc. We find that FGSM Acc is higher (lower) than Standard Acc when the step size is relatively large (small). The results suggest that NRF2 can still be exploited by an FGSM attack with a smaller step size to be *anti-correlated* with the true label. In other words, step size plays a non-trivial role when FGSM attack exploiting NRF2. The above results well explain why CO only occurs when the step size is set to a relatively large value (Wong et al., 2020).

Table 3: Evaluate FGSM Acc of CO model under different step sizes.

| step size (/255) | 0 | 1 | 2 | 3 | 4 | 5 | 6 | 7 | 8 |
|---|---|---|---|---|---|---|---|---|---|
| FGSM Acc (%) | 85.77 | 37.27 | 41.57 | 66.78 | 86.93 | 95.16 | 96.94 | 96.63 | 94.72 |

**A dynamic view on the CO from the NRF2 perspective.** Prior works analyzing CO mainly focus on Phenomenon 1 about low PGD Acc, which seems to be a pseudo-static state since the PGD Acc stays at zero after CO. Here, we investigate CO model further by analyzing FGSM Acc at different epochs, as shown in Figure 4. Fig. 4 show that the FGSM Acc under large step sizes consistently gets higher with more

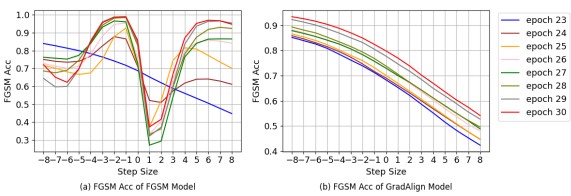

Figure 4: FGSM Acc with different step sizes. (a) FGSM AT where CO occurs at epoch 24; (b) GradAlign model where CO does not occur.

training epochs, suggesting the model continues to rely more on NRF2. In other words, CO can be perceived as a dynamic state of learning NRF2, which does not stop after the drop of PGD Acc. This is reasonable because NRF2 can be very useful features under FGSM attack.

## 4.2 CAN NRF2 ALSO JUSTIFY PHENOMENON 1?

The above analysis verifies the existence of NRF2 in a CO model, which well justifies the improved accuracy after FGSM attack (Phenomenon 2). Here, we discuss whether it can be used to justify Phenomenon 1. Regarding the relationship between NRF2 and PGD Acc, we formulate the following conjecture.

*Conjecture 2*: We conjecture that NRF2 can be a cause of a significant PGD Acc drop.

**Verification of Conjecture 2.** To verify Conjecture 2, we finetune a robust model on a training dataset with and without such NRF2, respectively, and evaluate the PGD Acc on the original test set with true labels ($X_{test}$, $y$). To minimize the influence of other NRF types, we adopt a model pretrained by PGD AT, which mainly has RFs, for the finetuning experiment. Specifically, we adopt the generated new training dataset ($X + \delta$, $y_{rand}$) at Step 2 of Fig. 2(c) as the one with NRF2. For the counterpart dataset without NRF2, we remove the added perturbation $\delta$, and thus ($X$, $y_{rand}$) is

used for training. The basic loss is set to cross-entropy (CE) to encourage learning the features, if any, in the generated dataset. However, the accuracy will quickly reduce to zero due to the random choice of $y_{rand}$. Thus, a KL loss, which encourages the output of the finetuned model to be close to that the pretrained mode, is added on top of the CE loss to encourage the model in the finetuning process to maintain the original RFs. The total loss is shown as follows:

$$Loss_{finetune} = CE(f(x + \delta; \theta), y_{rand}) + \lambda * KL(f(x + \delta; \theta_{pretrain}), f(x + \delta; \theta)), \quad (4)$$

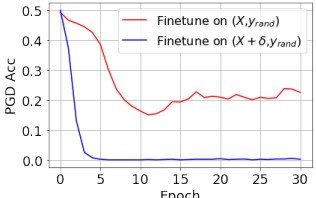

where $f(x; \theta_{pretrain})$ and $f(x; \theta)$ indicate the pretrained PGD model and finetuning model, respectively. A detailed setup for this experiment is reported in the Appendix. The results with $\lambda$ set to 5 are shown in Fig. 5. We observe that the PGD Acc can be maintained around 25% after 30 epochs of finetuning for the dataset $(X, y_{rand})$ which contains no features. By contrast, under the same setting, the PGD Acc quickly decreases to a value close to zero for the generated dataset $(X + \delta, y_{rand})$ which contains NRF2. The contrasting results verify the claim in Conjecture 2.

Figure 5: Finetuning a pretrained PGD AT model.

**Additional results with other $\lambda$ values** in Equation 4 are report in Fig. 6. As $\lambda$ gets larger, the model finetuned on $(X, y_{rand})$ maintains more RFs learned in pretrained weights $\theta_{pretrain}$, leading to an increase in accuracy. However, the PGD Acc for the model finetuned on $(X + \delta, y_{rand})$ (with NRF2) is zero for a wide range of $\lambda$ values, which is much lower than the model finetuned on $(X, y_{rand})$ (without NRF2). This further verifies the claim in Conjecture 2. Interestingly, the result in Fig. 6 can also be viewed as another proof for Conjecture 1. The Standard Acc, evaluated on the original test set $(X_{test}, y)$, is higher for the model finetuned on $(X + \delta, y_{rand})$ is higher than its counterpart on $(X, y_{rand})$ for all $\lambda$ in Fig. 6(b). This finding aligns with Conjecture 1 that there exists a type of NRF, which can be encouraged under FGSM attack.

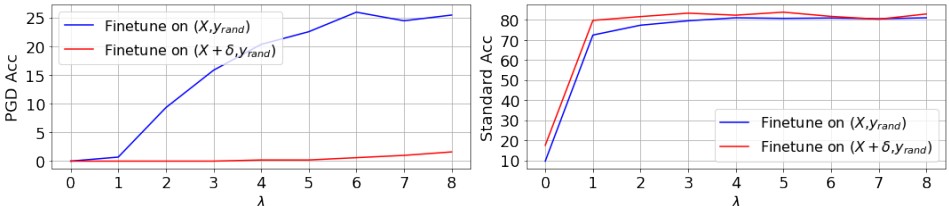

Figure 6: Accuracy of the finetuned model with different lambda. $\lambda=0$ indicates no KL term is introduced, and the model maintains more features of pretrained weights if $\lambda$ is larger.

**Discussion on the drop speed of PGD Acc from the NRF2 perspective.** As demonstrated in Section 4.1, CO can be perceived as a dynamic process of learning NRF2. With this insight, learning NRF2 in FGSM AT does not occur suddenly, which is supported by the finding that FGSM Acc still increases even after CO happens. If this is the case, how can we justify the sudden drop of PGD Acc within one epoch? At first sight, it seems that NRF2 can only explain the PGD Acc drop but not its drop speed. However, we argue that the sudden drop of PGD Acc is due to the worst-case property of PGD attack. Note that PGD attack seeks the most effective adversarial perturbation with multiple iterations to fool the model by exploiting the most vulnerable features in the model. In other words, the model is already vulnerable to PGD attack even if it only learns a small amount of NRF2 (one epoch regarding CO, for instance). After the PGD Acc drops to zero, the model continues to learn more NRF2, leading to a higher FGSM Acc.

## 5 NRF2 HELPS EXPLAIN HOW SOTA METHODS PREVENT CO

A recent work (Zhang et al., 2022) outperforms prior methods in FGSM AT by a large margin without additional computation overhead. Specifically, it shows that adding noise to the input (instead of initializing the adversarial perturbation with noise as in (Wong et al., 2020)) is critical for its success (Zhang et al., 2022). A similar finding has also been reported in another recent work (de Jorge

et al., 2022). However, why such a simple technique of adding noise on the images is so effective remains not fully clear. Here, we show that NRF2 sheds new light on their success.

Intuitively, the model tends to learn those features that are useful for prediction. Therefore, PGD AT mainly learns RFs because NRFs are not useful under PGD attack. With FGSM AT, the model is encouraged to learn NRF2 because FGSM attack increases its usefulness. Moreover, with our analysis in Section 4, CO can be seen as a dynamic process of learning NRF2. Therefore, the key to preventing CO in FGSM AT lies in decreasing the NRF2 usefulness under FGSM attack. Regarding why adding noise to the image input prevents CO, we establish the following hypothesis.

*Conjecture 3*: We conjecture that adding noise to the input decreases the usefulness of NRF2 under FGSM attack (indicated by FGSM Acc).

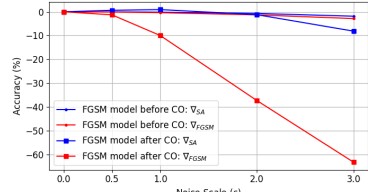

**Verification of Conjecture 3.** For facilitating the discussion, we divide all types of features into NRF2 and non-NRF2. A CO model has both NRF2 and non-NRF2, while a non-CO model mainly has non-NRF2. We evaluate the performance on a model without or with random noise added to the input and calculate the noise-induced change of Standard Acc and FGSM Acc (Table 4). Note that for FGSM Acc with noise, the noise is added to the input before the FGSM attack following (Zhang et al., 2022). For the model before CO, the noise has almost the same influence on the change of Standard Acc and FGSM Acc, *i.e* $\nabla_{SA}$ of $-0.70\%$ (Standard Acc change) is close to

Figure 7: Adding Uniform noise on input images during evaluation. Noise scale indicates the multiple of $\epsilon$.

$\nabla_{FGSM}$ of $-1.30\%$. We further conduct the same experiment on a CO model. Before adding noise, the FGSM Acc ($94.67\%$) is higher than its standard Acc ($85.77\%$), which can be attributed to NRF2 as in Conjecture 1. After adding noise, this trend is reversed ($57.37\% < 84.55\%$), suggesting Phenomenon 2 disappears in this setup. Moreover, $\nabla_{FGSM}$ ($-37.30\%$) is much more significant than $\nabla_{SA}$ ($-1.22\%$). Such a significant drop of FGSM Acc ($\nabla_{FGSM}$) on a CO model (with NRF2) suggests that the NRF2 usefulness under FGSM attack is significantly decreased. Fig. 7 visualizes $\nabla_{FGSM}$ and $\nabla_{SA}$ of different noise sizes, which shows the same trend with Table 4 that $\nabla_{FGSM}$ of CO model is the most significant change among all settings. Therefore, Conjecture 3 is verified, which provides a new understanding on why input noise prevents CO.

Table 4: Adding noise on the input images of FGSM AT model before and after CO **during evaluation**. $\nabla_{SA}$ and $\nabla_{FGSM}$ indicate the accuracy drop after adding Uniform noise, with noise scale $2\times\epsilon$ ($2\times8/255 = 16/255$).

| Evaluation model | NRF2 | non-NRF2 | Standard Acc | | | FGSM Acc | | | $\nabla_{FGSM}$ - $\nabla_{SA}$ |
| --- | --- | --- | --- | --- | --- | --- | --- | --- | --- |
| | | | original | with noise | $\nabla_{SA}$ | original | with noise | $\nabla_{FGSM}$ | |
| FGSM model before CO | No | Yes | 66.83 | 66.13 | -0.70 | 41.95 | 40.65 | -1.30 | -0.60 |
| FGSM model after CO | Yes | Yes | 85.77 | 84.55 | -1.22 | 94.67 | 57.37 | -37.30 | -36.08 |

**More discussion on NRF2 explaining earlier attempts of mitigating CO.** Even though we mainly apply our NRF2 to understanding the SOTA technique of input noise in recent works (Zhang et al., 2022; de Jorge et al., 2022), it also well justifies earlier successful attempts. For example, the success of random initialization in (Wong et al., 2020) is conceptually similar to adding the noise on the input but the noise magnitude is limited by the allowable perturbation size. (Kim et al., 2020) alleviates CO by limiting the step size, which aligns well with our finding in Table 3. (Li et al., 2020) avoids CO by switching to PGD AT after detecting the occurrence of CO, the success of which is expected since PGD attack can effectively discourage the model from learning NRF2.

## 6 CONCLUSION

The reason for CO in FGSM AT remains not fully clear despite various attempts to mitigate it. In contrast to prior works mainly studying PGD Acc drop to understand CO, our work focuses on another intriguing phenomenon that FGSM Acc is higher than Standard Acc. We have found that there exists NRF2 whose usefulness is decreased under FGSM attack and CO can be seen as a dynamic process of learning such a type of NRF. Our investigation has also provided a new understanding of successful attempts on how to mitigate CO in recent works.

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

# A  APPENDIX

**Experimental setups for NRF categorization in Fig. 2.** At Step 1, we follow the settings in (Andriushchenko & Flammarion, 2020), and train $\mathcal{M}_1$ on CIFAR10 for 30 epochs and cyclic learning rate with the maximum learning rate 0.3. Both attack radius sizes for training at Step 1 and perturbation generation at Step 2 are set as 8/255. Based on the new dataset $(X + \delta, y_{rand})$, $\mathcal{M}_2$ is trained for 30 epochs with a constant learning rate 0.015.

**Experimental setups for the finetuning experiment in Section 4.2.** The first two steps of the finetuning experiment follow the same settings of that in Fig. 2, generating a new dataset $(X + \delta, y_{rand})$. at Step 3 , we first follow the settings of PGD AT in (Andriushchenko & Flammarion, 2020) and train a robust $\mathcal{M}_{pgd}$. Based on the new dataset $(X + \delta, y_{rand})$, $\mathcal{M}_2$ is trained by finetuning on $\mathcal{M}_{pgd}$ for 30 epochs with a constant learning rate 0.005.

