# OpenReview forum: "Understanding Catastrophic Overfitting in Fast Adversarial Training From a Non-robust Feature Perspective"
_ICLR.cc/2023/Conference — Submitted to ICLR 2023_

### Official Review · Reviewer_LEgj · 2022-10-22

**Confidence:** 4
**Correctness:** 1
**Technical Novelty And Significance:** 2
**Empirical Novelty And Significance:** Not applicable
**Recommendation:** 3

**Clarity, Quality, Novelty And Reproducibility:**

The introduction and related work are well written, and the ideas of the paper that investigates CO from the perspective of NRF is novel.

**Strength And Weaknesses:**

+Provides phenomenon 2 that has been overlooked before: FGSM accuracy is higher than standard accuracy for a CO model.

+Extend the perspective of non-robust feature to FGSM AT to explain CO.

-The core of this paper is the direction-based categorization of NRF, say, NRF1 and NRF2. But the verification of NRF2 is not convincing. Table 2 shows that NRF2 exists after CO, and not before CO, which is obviously contradictory. As far as I know, non-robust features are inherent properties of the data (derived from patterns in the data distribution [1]) that do not change with the training process. For example, the semantic features are RF and the background is NRF, and we cannot say that there is no NRF in the data after PGD AT training. Based on this point, the subsequent opinions and analysis seem to be self-justifying and not convincing.

[1] Ilyas, Andrew, et al. "Adversarial examples are not bugs, they are features." Advances in neural information processing systems 32 (2019)


**Summary Of The Paper:**

This paper proposes a more fine-grained (direction based) classification of non-robust feature (NRF) to explain two phenomena of CO: 1) the PGD accuracy drops to a value close to zero when CO happens; 2) FGSM accuracy is higher than standard accuracy for a CO model. The authors think that there is an NRF with increasing usefulness after FGSM attack. Based on this, an explanation is provided for a simple technique for CO mitigating.

**Summary Of The Review:**

I appreciate the authors exploring CO from the perspective of NRF, but the main claim of the paper is questionable, so I am more inclined to reject.

---

### Official Review · Reviewer_ykUA · 2022-10-24

**Confidence:** 4
**Correctness:** 1
**Technical Novelty And Significance:** 1
**Empirical Novelty And Significance:** 1
**Recommendation:** 3

**Clarity, Quality, Novelty And Reproducibility:**

W1.	The existence of NRF2 is not justified because its relation with DNRF is unclear. I don't event understand why you name DNRF "double" as FGSM is clearly not the same as PGD with only one iteration. The latter considers gradient magnitude .
W2.	The proposed explanation lack justification. Many experiments only reproduce previous results without justifying that the reasons are due to the proposed conjecture. Merely showing gaps in Acc is not sufficient to explain that CO phenomena happen as the ways you thought.
W3.	The paper points out some ways to prevent CO, like adding noises. But there is no discussion on why adding noise prevents CO.


**Strength And Weaknesses:**

S1. The paper is well-written in general
S2. The idea of categorizing NRFs to understand CO is new


**Summary Of The Paper:**

Based on the finding of Non Robust Features (NRF) in the previous work (Ilyas et al., 2019), this paper tries to explain the phenomenon of  Catastrophic Overfitting (CO) in FGSM-based adversarial training. It categorizes NRFs into different types and tried to design experiments to show that the two commonly observed phenomena of CO, i.e., 1) the PGD Acc drops to a value close to zero when CO happens, and 2) FGSM Acc is higher than Standard Acc, is due to a specific type of NRFs.

**Summary Of The Review:**

The paper studied an interesting problem. However, the proposed conjectures lacks theoretical and empirical justifications, making them unconvincing.

First, the authors define DNRF as the features that are anti-correlated with the true labels by both FGSM and PGD attack. However, the authors did not explain why Figure2(b) can verify DNRF. It seems that the authors assume the features that are anti-correlated with the true labels by FGSM attack are the subset of the features that are anti-correlated with the true labels by PGD attack, but there’s no explanation or proof about this assumption. Note that “PGD attack is stronger than FGSM attack” does not explain the assumption. As far as I know, PGD is not the same as FGSM even with only one iteration. The relationship between DNRF and NRF2 is unclear to me. How do you make sure that NRF2 also exists in DNRF?

The existence of NRF2 also lacks justification. I don't understand why NRF2 can be exploited by Figure2(c) and have the following questions:
1) Why the perturbation of FGSM attack far/close from y_rand can increase/decrease the usefulness of NRF?
2) How to justify that the acc increment really comes from NRF2 instead of other features or side effects, such as the binary signing, in FGSM?
3) Why there’s no counterpart NRFs in PGD, i.e., why there’s no NRF whose acc is decreased/increased after PGD attack thus can be exploited by PGD attack to decrease/increase the acc after PGD attack?

There seems to be a contradiction between the definition of NRF2 and the experiments results shown in Table3, where the former defines NRF2 are the NRFs that the usefulness (correlation with true label, following Eq 2) increases while the latter concludes that NRF2 can be anti-correlated with true label under smaller step size.

There’s no evidence of the existence of NRF2. There’s also no visualization nor ground-truth injection of NRF2. Merely showing gaps in Acc is not sufficient to explain that CO phenomena happen as the ways you thought. Even though we assume that DNRF and NRF2 exist, they do not explain why these features increase only FGSM Acc but not Standard Acc, given that the FGSM Acc is known to be higher than Standard Acc when CO happens.

The experiment of verification of Conjecture 2 is also confusing. Why not just evaluate the PGD Acc variation with (X_train, y) and (X + δ, y) against a PGD AT model? If NRF2 really hurts the performance, (X + δ, y) should give worse Acc than (X_train, y). Why is the fine-tuning necessary?

Last but not the least, the authors tried to explain why adding noise prevents CO by “adding noise to the input decreases the usefulness of NRF2 under FGSM attack.” Again, there's no justification. I don't understand why adding noise to the input decreases the usefulness of NRF2 under FGSM attack. Your verification of Conjecture 3 is only a reproduction of (Zhanget al., 2022)'s experiments. And Figure 7 only shows that adding noise can prevent CO, which has nothing to do with NRF2.

---

### Official Review · Reviewer_x5Nw · 2022-10-24

**Confidence:** 4
**Correctness:** 3
**Technical Novelty And Significance:** 2
**Empirical Novelty And Significance:** 2
**Recommendation:** 3

**Clarity, Quality, Novelty And Reproducibility:**

This paper is generally organized well but its clarity needs to be improved. For example, the robust/non-robust features are defined in Section 3.1, but are never used in the rest of the paper. Instead of plain words, it would be better if the authors can extend the definitions to abstract the claims in a rigorous manner.

The quality of the paper relies on the correctness and comprehensiveness of the experiments, given only empirical evidences are presented in the paper. I believe experiments with more settings can help, for example, different model architecture and datatsets. But it would be best if the authors can work on a toy datasets (e.g. in [2]) and provide some theoretical justification.

The novelty of this paper is mostly attributed to the new observations about the CO phenomnenon. The focus on the specicial property of CO (phenomenon 2 as mentioned in this paper) itself is not novel since it has been discussed in previous works [1].

[1] Adversarial Machine Learning At Scale. Kurakin et al.

[2] Adversarial Examples Are Not Bugs, They Are Features. Ilyas et al.


**Strength And Weaknesses:**

This paper is well-organized and presents the problem clearly. It focuses on an important problem in adversarially robust learning, and made interesting observations.

My biggest concern of this paper is that it might miss an important citation. [1] has discussed in detail about the phenomenon studied in this paper, namely accuracy against FGSM adversarial examples are higher than standard accuracy. In their paper,  the phenomenon is dubbed as the "label leaking" effect. They provide a very simple explanation to this phenomenon. One-step attack methods that use the true label perform a very predictable transformation that the model can learn to recognize. The generated adversarial examples thus may inadvertently leak information about the true label. I believe this explanation is not discussed anywhere in this paper. Furthermore, such a simple explanation and the observations made in this paper are not contradictory. In fact, the observations in this paper can further support the 'label leaking' explanation.

Nevertheless, some observations made in this paper are indeed novel and interesting. For example, in section 4.1, at "verification of conjecture 1", the authors verified the existence of a particular type of non-robust feature (NRF2) by showing the adversarial examples generated by a CO model can be learned by another model and achieve non-trivial standard accuracy. Compared to the original CO observation, this further shows that the information about the true label leaked in the adversarial example is in fact transferable, which is worth further investigation.

[1] Adversarial Machine Learning At Scale. Kurakin et al.



**Summary Of The Paper:**

This paper studies the phenomenon of catastrophic overfitting (CO) in adversarially robust training with a single step perturbation method FGSM. It brings ones's attention to an intriguing property of CO, namely the accuracy on FGSM attacked samples is higher than that on clean samples. The authors try to explain this phenomenon from a robust/non-robust feature perspective. They conjecture the existence of a special type of non-robust feature that is particularly useful to the classification, and verify it based on controlled experiments. They also show that the non-robust feature perspective can potentially explain the recent practice on combating CO, namely adding random noise on the inputs.

**Summary Of The Review:**

This paper made some interesting observations about catastrophic overfitting in adversariral training. But its contribution is hard to measure without comparative discussion with previous work studying this phenomenon. The clarity and quality also need to be improved as mentioned above.

---

### Official Review · Reviewer_xEW8 · 2022-10-25

**Confidence:** 5
**Correctness:** 2
**Technical Novelty And Significance:** 3
**Empirical Novelty And Significance:** 3
**Recommendation:** 5

**Clarity, Quality, Novelty And Reproducibility:**

**Clarity**: The paper is hard to read.

**Quality** The idea is compelling and valuable, but the experiments could be cleaned up.

**Novelty** I believe this work is novel.

**Reproducibility** The main experiments are probably reproducible, However, the main findings are only supported by weak evidence on a single dataset.

**Strength And Weaknesses:**

# Strengths

1. **Interesting idea**: The idea that CO could be caused by the existence of certain kinds of non-robust features in the data is very compelling and valuable.
2. **Good empirical methodology** I really appreciate the presentation of the experiments in this work in which each experiment is motivated by a research question and a plausible hypothesis.
3. **Interesting insights for fast AT methods** personally, I believe that the insights in Sec. 5 can be relevant to the community as they give more information as to why fast AT methods based on noise injection work.

# Weaknesses

1. **Only partial and weak evidence** One of the main weaknesses of this work is that the main experiment in Sec 3 only provides some weak signal in support of the main hypothesis. 17% accuracy is indeed slightly higher than trivial accuracy, but it is significantly lower than the original 48% used by Ilyas et al. to corroborate their conjecture. In this regard, ss the hypothesis presented in this work is rather complex, one would expect to have more convincing evidence that it is true in all settings. Something that would alleviate my concerns would be seeing a replication of this experiment on other datasets (e.g. SVHN, CIFAR100, ImageNet100…) and other models which showed that the weak numbers observed for CIFAR10 are at least consistent in other settings.
2. **Poor clarity** In general, I have found the central sections of this work very hard to read. The constant use of acronyms, references to observations as non-descriptive Phenomenon 1 or 2, and in general, the convoluted writing to describe an already-complex hypothesis, make this paper very hard to read.
3. **Unpolished experiments** I find some experimental setups in the text a bit strange and full of unjustified moving pieces that just add further complexity to test complex conjectures. For example, I am not sure I understand why the regularisation term of Eq. 4 is fully necessary, or why as robust model is needed to verify conjecture 2.
4. **Some speculation** While I certainly believe this work provides partial evidence to connect CO with the features of the data, I believe it does not do the same to explain the sudden onset of CO in fast AT. The explanations of the authors in this regard are not substantiated by clear evidence and, thus, remain speculations in my mind.
5. **No clear refutation of alternative hypothesis** Although in their introduction, the authors argue that the non-linear hypothesis of Andriushchenko & Flammarion cannot explain Phenomenon 2, there is no clear evidence in this work that explains why it is wrong. There could be other additional hypothesis on top of their non-linear conjecture that could explain that phenomenon, However, the NRF hypothesis of this work does fail to explain the key observations of Andriushchenko & Flammarion which is that after CO the models become highly non-linear. In the view of the authors, how does the NRF theory then explain this phenomenon, and why does GradAlign succeed in preventing CO?

**Summary Of The Paper:**

This paper proposes a new explanation for catastrophic overfitting based on the existence of a special type of non-robust features which FGSM makes easier to learn. To demonstrate this, the authors reuse the experiments of (lyas et al. 2019) to validate their conjecture by showing that a training on a dataset containing only this type of non-robust features yields non-trivial accuracy. They later discuss how their new hypothesis can explain other observations regarding CO with special emphasis in explaining the success of injecting noise during fast-AT. Their working hypothesis, which they validate on some experiments, is that adding noise makes learning the non-robust features harder.

**Summary Of The Review:**

Overall, I believe this is a borderline paper. The main hypothesis has merit and it is an interesting contribution to the community. However, this work only partially corroborates its truth based on weak evidence on a single dataset, and fails to explain prior observations. The text is  not easy to read, and some experiments have too many complicated design choices that do not manage to isolate the right effects.

At this stage, I am leaning towards rejection, but if the authors can convincingly show evidence that their findings can be replicated in other contexts and clarify my concerns of **weakness 5**, I might be willing to increase my score and lean to accept.

---

### Decision · Program_Chairs · 2023-01-20

**Decision:**

Reject

**Justification For Why Not Higher Score:**

The current paper does not seem to be easily improved by superficial modifications. The weaknesses of this paper make it unacceptable for the ICLR conference. All reviewers are negative about it. The authors of this paper did not provide responses to the concerns of reviewers. The area chair hence recommends rejecting this paper.

**Justification For Why Not Lower Score:**

N/A

**Metareview: Summary, Strengths And Weaknesses:**

This paper presents a new perspective on catastrophic overfitting in adversarial training. It reuses the experiments of prior work to justify that training on data containing only non-robust features yields non-trivial accuracy. Besides, this paper shows that the perspective of non-robust features can potentially explain the recent practice of handling catastrophic overfitting.

The strength of this paper lies in: the research problem is interesting. The idea that catastrophic overfitting can be caused by non-robust features is insightful to the research community.

The weaknesses of this paper are summarized as:

(1) This paper misses the discussions about related literature [R1], where a similar experimental phenomenon was observed. The absence of this discussion makes the contribution of the paper unclear.

(2) The verification of NRF2 is not convincing. Additionally, there is no complete explanation of the presence of non-robust features.

(3) There are not sufficient justifications and explanations for empirical results. The results cannot well support the alternative hypothesis sometimes. Experimental settings are not comprehensive enough. More experiments, e.g., with more network structures and benchmarks could be helpful.

(4) The writing should be more logical, Section 3 in particular. Reviewers and AC found it somewhat difficult to follow.

[R1] Alexey Kurakin et al. Adversarial Machine Learning at Scale. arXiv:1611.01236